# Exceptionally High Cystic Fibrosis-Related Morbidity and Mortality in Infants and Young Children in India: The Need for Newborn Screening and CF-Specific Capacity Building

**DOI:** 10.3390/ijns11030067

**Published:** 2025-08-22

**Authors:** Priyanka Medhi, Grace R. Paul, Madhan Kumar, Grace Rebekah, Philip M. Farrell, Jolly Chandran, Rekha Aaron, Aaron Chapla, Sneha D. Varkki

**Affiliations:** 1Department of Paediatrics, Christian Medical College, Vellore 632004, India; child3@cmcvellore.ac.in (P.M.); dr_madhankumar@yahoo.com (M.K.); 2Division of Pulmonary and Sleep Medicine, Nationwide Children’s Hospital, Columbus, OH 43205, USA; grace.paul@nationwidechildrens.org; 3Department of Biostatistics, Christian Medical College, Vellore 632004, India; gracesamuel@cmcvellore.ac.in; 4Departments of Pediatrics and Population Health Sciences, University of Wisconsin School of Medicine and Public Health, 600 Highland Avenue, Madison, WI 53792, USA; pmfarrell@wisc.edu; 5Paediatric Critical Care Unit, Christian Medical College, Vellore 632004, India; picu@cmcvellore.ac.in; 6Molecular Genetics Laboratory, Department of Medical Genetics, Christian Medical College, Vellore 632004, India; rekha.athiyarath@gmail.com; 7Molecular Endocrinology Laboratory, Christian Medical College, Vellore 632004, India; aaronchapla@gmail.com

**Keywords:** India, cystic fibrosis, newborn screening, mortality, anemia, edema, incidence

## Abstract

Early diagnosis of cystic fibrosis (CF) through newborn screening (NBS) improves clinical outcomes, but in countries like India, delayed diagnosis increases morbidity, mortality, and likely underestimates infant deaths from CF. We performed a retrospective study at a single center in south India from 2017 to 2025 reviewing children diagnosed with CF before one year of age. Patient demographic, clinical, and genetic data were analyzed to characterize early clinical features and identify factors linked to mortality. Of 56 infants diagnosed with CF, 59% survived (median current age 55 months) while 41% died (median age of death 5 months). Key clinical indicators included sibling death with CF-like symptoms, rapid weight loss, and persistent respiratory or nutritional complications. Mortality risk under one year was significantly linked to hypoalbuminemia (OR 9.7), severe malnutrition (OR 4.4), severe anemia (hemoglobin < 7 g/dL) requiring blood transfusions (OR 3.0), and peripheral edema (OR 4.2). A triad of anemia, hypoalbuminemia, and edema was found to strongly predict death (OR 4.2). Integrating clinical checklists of these manifestations into primary healthcare may improve prompt referrals for earlier diagnosis and treatment. Continued education and advocacy for NBS are essential to reduce potentially preventable CF-related deaths in young children.

## 1. Introduction—Cystic Fibrosis in India

Data on the incidence of CF in India are sparse. Farrell et al. [1] summarized estimations ranging from 1:7000 to 1:12,000. The estimated prevalence of CF, extrapolated from immigrant populations in the USA and UK, ranges from 1 in 10,000 to 1 in 40,000 [2]. In India, the median age of CF diagnosis varies between 2 years and 10.5 years (based on the cohort reported), and is notably delayed compared to the USA (2.6 months) and Europe (3.6 months), indicating that many infants with CF are not diagnosed nor initiated on CF specific treatment in the critical period of early infancy and consequently many may succumb to the disease [3,4,5,6]. Also, the carrier rate of any pathogenic *CFTR* variant was reported at 4.5% in a North Indian cohort which is quite alarming in a nation with relatively high consanguinity rates [7]. With approximately 24 million births annually, this indicates a potentially significant burden of undiagnosed cases. Though never systematically addressed, reports suggest that infant and early-childhood mortality rates among children with CF in India are grossly underestimated. In the absence of standardized newborn screening (NBS), it is essential to enhance our understanding of the early clinical presentation of CF in infants and refine diagnostic approaches that are feasible in resource-limited settings.

Our study describes the demographic and clinical profile of children diagnosed with CF in infancy at a single center in south India where NBS is not available and examines clinical and survival outcomes to identify risk factors associated with early mortality. We recognize that there are characteristic clinical manifestations (“red flags”) that should alert clinicians to a high probability of life-threatening CF and that some signs should be regarded as severe enough to require urgent intervention. We addressed these questions: If NBS cannot be immediately implemented to identify all babies affected by CF, what can be done now?Can we diagnose CF reliably in the absence of NBS?Can we create a simple clinical screening tool to identify those who are at risk of dying in the first few months of life and focus our attention on them?Can we integrate such clinical tools during routine healthcare for infants?

## 2. Methods

### 2.1. Location

This retrospective study was conducted with Institutional Review Board (IRB Min No 2412130 dated 18 December 2024) approval at Christian Medical College (CMC), Vellore, a tertiary care referral center in the south India state of Tamil Nadu serving patients referred from southern and eastern regions of India. Patients registered in the CF clinic’s database from January 2017 to January 2025 were included.

### 2.2. Subjects and Diagnostic Confirmation

Children with CF were included if the diagnosis was established before one year of age, based on suggestive clinical features along with elevated sweat chloride concentrations (≥60 mmol/L using Wescor Macroduct^®^ (ELITechGroup Inc., Logan, UT, USA) sweat collection and chloride analysis) and/or the identification of two biallelic pathogenic or likely pathogenic *CFTR* gene variants [8]. In addition, three infants with classic CF phenotype were included post-mortem when their respective siblings were later confirmed to have CF by genetic analysis along with parental validation of carrier status. All infants in the study period were included. *CFTR* mutation analysis was performed at our institution by next generation sequencing (NGS) of extracted DNA. The *CFTR* primer pool included the detection of intronic variants and deletion. Variants were classified per the American College of Medical Genetics guidelines [9]. Patients were classified as pancreatic insufficient when fecal elastase was <200 µg/gm of stool. In the absence of fecal elastase levels, infants were categorized as pancreatic insufficient if they had classic symptoms of steatorrhea and also had two *CFTR* variants associated with pancreatic insufficiency [8].

### 2.3. Data Collection

Patient demographics, clinical features, and genotypes were reviewed. Microbiological data were obtained from airway samples processed according to established CF culture protocols [10], and imaging data were primarily sourced from chest radiographs, which were analyzed for characteristic CF changes. Data were collected to monitor clinical outcomes (pulmonary and nutritional morbidity, and death) from time of diagnosis until January 2025. Factors impacting mortality and survival were reviewed for all 56 infants in this cohort.

### 2.4. Statistical Analysis

Data were systematically recorded in a structured format, and all data were de-identified prior to analysis. Descriptive statistics were employed to report the clinical profile of infants with CF in our cohort, with continuous variables presented as means with standard deviations or medians with ranges, and categorical data presented as frequencies and percentages. Comparative analyses were conducted to discern differences in clinical characteristics between infants who survived and those who did not. Statistical methods, including chi-square tests, *t*-tests, and regression analyses, were tailored to identify associations between clinical features and outcomes among CF infants. Statistical significance was defined as a *p*-value < 0.05, with analyses performed using SPSS version 21.0.

## 3. Results

### 3.1. Study Cohort

During the study period, 128 children below the age of 18 years, including 56 infants, were registered in the CF clinic. Of the 128 children, 87 are alive, and 41 have died. In this study, we focused on the 56 children who were diagnosed as having CF in infancy, and of them, 33 (59%) are alive and 23 (41%) have died as of January 2025. The median current age of those alive was 55 months (range 7–110 months), and the median age of death of the 23 infants was 5 months (range 2–87 months).

In the subsections below, we report the clinical profiles of infants with CF to identify features that might be valuable to achieve early clinical diagnosis and report the characteristic demographic and clinical manifestations that may signal and predict early mortality, thereby promoting awareness on potentially preventable or treatable morbidity and mortality among infants.

### 3.2. Clinical Profiles of 56 Children Diagnosed with CF in Infancy

Of the 56 infants with CF, 38 were male and 18 were female, and 94% were born at term, with 25% having a low birth weight (<2.5 kg). Table 1 describes the clinical profile of the 56 children diagnosed in infancy. Genetic data by NGS of the *CFTR* gene were available for 55 patients. Class I or II *CFTR* mutations were identified in 62.2% of cases. Homozygous *CFTR* mutations were observed in 58.2% of infants. A detailed distribution of the alleles identified is presented in Figure 1. Fecal elastase concentrations were available for 51 of the 56 patients, with 44 confirmatory for pancreatic insufficiency. Five patients with history of steatorrhea were classified as being pancreatic insufficient based on the presence of a biallelic homozygous or compound heterozygous *CFTR* variant associated with pancreatic insufficiency (c.1521_1523delCTT, c.4231C>T, c.1029del, c.223C>T, c.274-1G>A) [8]. At least one sputum or deep throat swab culture was performed in 47 patients. Among them, 33 patients had positive cultures, while 14 showed only normal respiratory flora. *Pseudomonas* species were the most frequently isolated pathogen, detected in 28 of the 33 positive cases on at least one occasion, followed by *Staphylococcus aureus* in 10 cases and *Stenotrophomonas* in 2 cases. At the time of this review, abnormal chest radiographic findings were documented in 44 patients, with varying lobar involvement. Notably, 10.7% of them have established bronchiectasis by 5 years of age according to computed tomography imaging of the chest. Recurrent lower respiratory tract infections requiring oral or intravenous antibiotics were documented in 75% of cases, with an average of four episodes per year.

Of the 33 children who are alive at the time of reporting, 24 remain under regular follow-up care, with a median follow-up duration of 31 months (range: 1 to 86 months). Out of 27 children who reached 2 years of age, the most recent recorded BMI of 16 of them was 13.8 ± 1.8 kg/m^2^. Spirometry was documented in 3 of the 11 children who are above 6 years of age currently. Mean FEV1%pred was 75.3 ± 40.3%, with mean FVC%pred 82 ± 39.6% and FEV1/FVC ratio 76.3 ± 9.8.

### 3.3. Comparative Analysis Between Infants with CF Who Survived and Those Who Died

Table 2 describes the clinical profile and statistically significant variables that predicted survival versus mortality. As expected at this age, the key predictors of mortality were related to malabsorption and nutrition. Survival outcomes differed between pancreatic insufficient (PI) and sufficient groups (*p* = 0.018). All deaths occurred in the PI group, highlighting poorer outcomes among patients with intestinal malabsorption. Low serum albumin levels (*p* = 0.001), severe malnutrition with edema (*p* = 0.008), and need for blood transfusion (*p* = 0.001) were significantly associated with high mortality rate. Among the deceased, 70% had recurrent lower respiratory tract infections requiring hospitalizations, and 50% had at least one ICU admission. There were no statistically significant differences based on gestational age, or history of meconium ileus.

### 3.4. Risk Factor Analysis for Infants Who Died Prior to One-Year of Age

Table 3 describes univariate risk factor analysis for those who died before 1 year of age. The odds of under-1 year mortality were significantly higher if there was evidence of severe hypoalbuminemia < 2 gm/dL (OR 9.7, 2.1–44.6), features of severe malnutrition (OR 4.4, 1.2–12.2), a history of blood transfusion for severe anemia (hemoglobin < 7 gm/dL) (OR 3.0, 0.9–9.6), and peripheral edema (OR 4.2, 1.2–15.2). A clinical triad of severe anemia, severe hypoalbuminemia and edema was a significant predictor of mortality (*p* < 0.001, OR 4.2, 1.19–15.26).

## 4. Discussion

### 4.1. Significance of the Quantitative Assessments of Infant Growth in the Study Cohort

Our observations augment the emerging pool of evidence on the existence and potential severity of CF in India, specifically among infants and young children [12,13,14]. Apart from classic CF symptoms, family history of presumed or confirmed CF, along with an unexpected rapid decline in nutritional health, were key indicators to suggest CF. In our study cohort, infants with CF had a normal weight-for-age (WFA) z-score at birth but experienced a steep decline in the WFA z-score by six weeks of age as their protein-energy malnutrition progressed. The WFA z-score decreased from −1.76 to −3.9 among the infants who died, and from −1.43 to −3.13 among those who survived. These data reflect the challenge of achieving the very high growth velocity of infancy and add to prior evidence in which failure to thrive was documented as early as 6 weeks of age in 93% of untreated infants with CF, when WFA z-score decreased from −1.25 at birth to −4.22 at 6–8 weeks of age [15].

### 4.2. The Deadly Triad

It was alarming to note how many infants with CF needed hospitalization with severe signs and symptoms as early as 1 month of age and that 10 (20%) died before reaching 4 months of age. Perhaps most significantly, our data support the conclusion that several key clinical characteristics—most of which are readily identifiable—can effectively identify infants at risk of dying. A history of sibling death due to symptoms suggestive of CF, a steep decline in WFA z-score, and persistent respiratory or nutritional symptoms should instigate prompt diagnostic evaluation and treatment as indicated for a medical urgency. We wish to emphasize that the odds of mortality under 1 year of age were significantly higher if there was evidence of hypoalbuminemia (OR 9.7, 2.1–44.6), profound protein-energy malnutrition (OR 4.4, 1.2–12.2), a history of blood transfusion for severe anemia (hemoglobin < 7 gm/dL) (OR 3.0, 0.9–9.6), and peripheral edema (OR 4.2, 1.2–15.2). The clinical triad of severe anemia, hypoalbuminemia and edema was a significant predictor of mortality (*p* < 0.001; OR 4.2, 1.19–15.26). Although this combination was described [16] in Danish infants born during 1949–1980, its significance for predicting mortality has not been systematically analyzed. Consequently, our hypothesis has been confirmed that there are indeed characteristic clinical manifestations that should alert clinicians as “red flags” for a high probability of life-threatening CF and that some signs should be regarded as severe enough to require urgent intervention.

### 4.3. Severe Anemia—An Ominous Sign

The risk and impact of severe anemia were especially impressive and have been reported by others as relevant to the importance of early diagnosis through NBS [17]. Occurring so severely at such a young age without blood loss, the anemia is likely attributable to a hemolytic phenomenon caused by profound vitamin E deficiency as Wilfond et al. [17] reported after it had been proven that tocopherol deficiency can cause hemolysis in vitro and in vivo [18]. Most of the pancreatic-insufficient infants with anemia who died in our cohort did not even have an opportunity to receive any therapy for 2 weeks. This unfortunate situation calls for empiric nutritional and GI treatment with pancreatic enzymes at the first suspicion of CF in an infant with severe malnutrition and edema, severe anemia, and a history of sibling death. The same rationale and urgency applies to immediate treatment with alpha-tocopherol at a dose of 50 IU/day as recommended by Wilfond et al. [17]. Both pancreatic enzyme replacement therapy and vitamin E in proper doses are well tolerated without risks, whereas continuation of these nutritional complications in vulnerable infants can be fatal.

### 4.4. Dehydration and PseudoBartter Syndrome

PseudoBartter syndrome is not uncommon in India [19]. The strategy of proactively identifying babies who present with dehydration in summer resulted in better survival of those babies. This may be related to a milder phenotype and pancreatic-sufficient status. During extreme heat, the presentation of dehydration with PseudoBartter syndrome can be an indicator of CF, even in the absence of typical features related to fat malabsorption, because electrolyte losses occur independent of nutritional deficits. Fortunately, all these serious consequences of delayed CF diagnoses can be prevented by early diagnosis through NBS and rapid implementation of effective therapies.

### 4.5. Systematic Disease Recognition Tools

With the recognition that CF is a global disease [20,21,22], it is important for the low- and middle-income countries (LMICs) that do not yet have NBS to develop and apply systematic clinical recognition mechanisms as European and North American programs did many decades ago. We believe that our observations provide a pathway for a clinical tool that can be used for referrals to establish CF diagnoses and implement therapy quickly. Because Indian infants are routinely evaluated at 6–8 weeks in well-established immunization clinics, primary care providers in both the private and public health sectors have a special opportunity to aid in CF diagnoses. Our data support the imperative need for education and awareness of CF among community caregivers and general pediatricians, emphasizing careful attention to poor weight gain and the above-mentioned signs/symptoms. A simple checklist including clinical metrics such as failure to thrive, peripheral edema, summer dehydration, recurrent pneumonia along with laboratory-based alerts such as severe anemia (hemoglobin < 7 g/dL), hypoalbuminemia (serum albumin < 2.0 gm/dL), and dyselectrolytemia suggesting PseudoBartter syndrome should trigger an urgent referral for CF testing and maybe even empiric initiation of CF therapies (pancreatic enzymes, salt supplementation, fat soluble vitamins nutrition, airway clearance, antibiotics, supportive care, hydration, etc.) before these infants succumb to untreated disease. The recently developed CF-Clinical Diagnostic Scoring system described by Dhochak et al. [23] revealed a high diagnostic accuracy with good sensitivity and specificity, and underscores the value of systematic tools for clinical diagnosis when resources are limited.

### 4.6. Misdiagnosis and Missed Diagnosis

The overlap in clinical presentation of CF with prevalent endemic childhood illnesses (diarrheal diseases, malnutrition, recurrent pneumonia, etc.) increases the risk of CF being concealed within the broader categories of infant mortality. As a result, CF-related deaths may not be distinctly reported or recognized in national health statistics, similar to reports from the African continent [24]. In a previous study conducted at our pediatric ICU, we found that 7% of infant deaths were attributable to undiagnosed CF. This was determined by *CFTR* mutation analysis on stored blood samples collected from critically ill infants aged 1 to 6 months who were selected based on clinical suspicion [13].

Underdiagnosis is further confounded by the absence of routine CF NBS in India and limited awareness of the disease among healthcare providers. Routine implementation of CF NBS programs in India presents significant challenges due to under-recognition of the disease, resource limitations, scarcity of specialized CF care centers and laboratories, heterogeneity in the Indian population, and the influence of social determinants of health [25]. Therefore, it is imperative to strengthen epidemiological surveillance, increase diagnostic capacity, and integrate CF awareness into existing child health programs to better understand and address the hidden burden of CF in infancy across India. Underdiagnosis compounded with the consequences of delayed intervention, the limited availability of CF-specific personnel and therapies, and inadequate medical infrastructure to manage CF, are the primary reasons for high morbidity and mortality in India [13,15,20].

### 4.7. Newborn Screening—An Important Goal

There are multiple emerging initiatives towards NBS in India, with interest and support from both the government and private sectors. Most of these initiatives advocate for a multi-disease screening panel which would be the most effective, and cost-effective for India. In 2024, the Pan-Asian Kathmandu Declaration regarding NBS supported (1) collaborative educational ventures, (2) population studies, (3) quality control, (4) advocacy to policymakers, and (5) improved regional health infrastructure [26]. Our data reveal a high CF-related morbidity and mortality, especially in young children, which we believe are contributing to India’s high infant and under-5 mortality rate [13,27,28]. In a recent Commentary by Farrell et al. [1], the authors provide rationale to improve resource allocation for CF NBS by providing more evidence of the disease burden (which our data contribute to), and also advocate for formal NBS initiated as institution-based or district-based ventures. We believe that this is the time to collaborate and act!

Despite this awareness, the implementation of universal NBS throughout India remains a distant goal that is likely to be achieved by incremental progress on a regional basis as in Europe and the USA, but increasing advocacy efforts towards early diagnosis have helped initiate a few preliminary private and public sector ventures towards institution-based CF newborn screening. The Cystic Fibrosis Foundation is very supportive of CF NBS initiatives in LMICs, including India. Until NBS is standardized, our proposed clinical tool may facilitate recognition of life-threatening CF, and continued use after NBS implementation may help recognize critically ill infants with false negative screening results. NBS nationwide initiatives and policy changes depend on further validated regional/national evidence of disease prevalence and confirmation of our data reported herein on the mortality risk of CF, especially during infancy.

### 4.8. Limitations

Our study is limited by the retrospective nature of data collection and analysis. Because our observations are for a single center, our data may not be completely generalizable due to regional differences in population, resources, and medical infrastructure.

## 5. Future Directions

Considering the population size and varying medical infrastructure in India, education and awareness of CF will always be crucial and must be continuous. Our data emphasize the need for education on the urgency of diagnosing CF in early infancy and also should alert primary care providers to key clinical and laboratory parameters that warrant prompt referral for CF diagnosis and immediate treatment. One of our goals has been to create simple checklists as noted above and increasing awareness of CF by direct community outreach and educational campaigns at our affiliated and regional community medical centers where most babies are followed for routine immunizations. On a larger scale, we plan to include CF-related infant mortality in emerging registry systems and advocate that CF would be added as a distinct etiology for infant and early-childhood mortality in the government’s health sector database. This would be crucial to advocate for CF-specific therapies to be available, accessible, and affordable in India, when evidence on CF as a contributor to India’s high infant mortality rate is further validated. More data are required to augment this evidence, and we encourage CF providers dispersed across India to contribute to the emerging CF registry. Also, research initiatives to study the feasibility and sustainability of CF NBS will be crucial in the future. Although CF NBS is important, further investments into education and awareness, advocacy, training of CF-specific personnel, improving access and affordability for diagnosis and treatment at more specialized CF centers, and quality control, are necessary for successful and sustained clinical outcomes and survival.

## 6. Conclusions

This first study in India on a cohort of infants diagnosed to have CF within the first year of life provides strong evidence for giving high priority to this life-threatening genetic disease. Our data support education and awareness of CF to improve early neonatal diagnosis, and also describe key risk factors that predict high mortality despite relatively early diagnosis. Simple clinical checklists can be seamlessly integrated into existing well-established primary health sectors to refer patients for further diagnostic confirmation and early treatment initiation. Ongoing education and advocacy for early diagnosis by NBS remain imperative.

## Figures and Tables

**Figure 1 IJNS-11-00067-f001:**
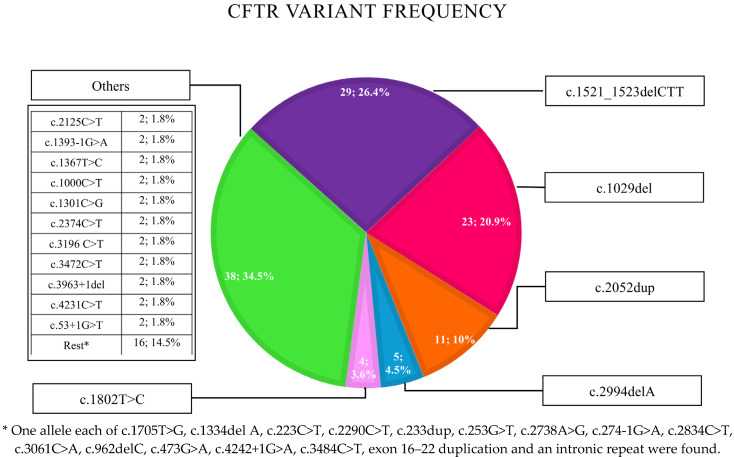
Graphical representation of *CFTR* variants with allele frequency of 55 infants with CF.

**Table 1 IJNS-11-00067-t001:** Demographic, clinical, and laboratory parameters of 56 children diagnosed in infancy.

Demographics	Results
Median current age of those alive (months) *n* = 33	52.4 (7–110 months)
Age breakdown	
- <1 year	4
- 1–5 years	15
- >5 year	14
Median age of death of non-survivors (months) *n* = 23	5 (2–87 months)
Age breakdown	
- <1 year	18
- 1–5 years	4
- >5 years	1
Median age at Diagnosis (months) *n* = 56	5.4 (1–11) months
Male:Female	2:1
Consanguineous parentage (*N* = 55 families)	19 (33.9%)
**Clinical Features at Diagnosis**	**Numbers (%)**
Term Birth (Gestational age ≥ 37 weeks)	53 (93.6%)
Low Birth weight (≤2500 g)	14 (25.0%)
Meconium ileus	7 (12.5%)
Exocrine pancreatic insufficiency *	49 (87.5%)
PseudoBartter presentation	14 (25.0%)
Hypoalbuminemia < 2 gm/dL	11 (20.7%)
Severe malnutrition with edema ** at presentation	15 (28.6%)
Recurrent pneumonia prior to diagnosis ^#^	42 (75.0%)
Severe Anemia needing blood transfusion ^##^	24 (42.8%)
**Investigations**	
Sweat Chloride ≥60 mmol/L	30 (81%)
30–59 mmol/L	4 (10.8%)
<30 mmol/L	3 (8.1%)
Not available	19
**Zygosity of Common *CFTR* Variants**	
- Patients with at least one F508del variant	20/55 (36.4%)
- Patients homozygous for F508del variant	9/55 (16.4%)
- Patients homozygous for c.1029del variant	8/55 (14.5%)
- Patients with other mutations ^	18/55 (32.7%)
- No data	1/56
**Mortality Related to CF**	
No. of families who had already lost a child to confirmed CF/probable CF	15 (26.8%)
Mortality in the cohort	23/56 (41.0%)

* Fecal elastase level of less than 200 µg/g of stool or by identifying two biallelic pathogenic *CFTR* variants associated with pancreatic insufficiency. ** Weight-for-length below −3 SD with the presence of edema. ^#^ More than three separate episodes requiring antibiotics [11]. ^##^ Severe anemia (hemoglobin < 7 g/dL) needing blood transfusion before diagnosis due to ongoing respiratory distress/prolonged oxygen requirement. ^ See Figure 1.

**Table 2 IJNS-11-00067-t002:** Comparison of clinical and demographic variables between patients who survived and those who did not (Total *n* = 56).

Variables	Patients Who Died (*n* = 23, 41%)	Patients Who Survived (*n* = 33, 59%)	*p* Value
Median current age of those alive (months)	N/A	55 (7–110) months	N/A
Median age of death of non-survivors (months)	5 (2–87 months)	N/A	N/A
Mean age at diagnosis (months)	4.9 (±2.5)	5.7 (±3.1)	0.28
Presence of homozygous CFTR variants. (Most common: F508del, c.1029del)	19 (82.6)	13 (40.6)	<0.002
Mean weight at birth (gm)	2590 (±560)	2770 (±390)	0.16
Weight z-score at birth	−1.726	−1.43	0.39
Mean weight at 6–8 weeks (gm)	2700 (±670)	3500 (±750)	0.002
Weight for age z-score at 6–8 weeks	−3.908	−3.136	0.22
Decrease in weight z-score from birth to 6–8 weeks	−2.33	−1.75	0.37
Exocrine pancreatic insufficiency * (%)	23 (100)	26 (78.8)	<0.034
Hypoalbuminemia (<2 gm/dL) (*n*, %)	10 (43.5)	1 (3)	<0.0001
Severe malnutrition with edema ** (n, %)	11 (47.8)	5 (15.2)	<0.008
Severe anemia requiring blood transfusion before 6 months of age ^#^ (*n*, %)	16 (69.6)	8 (24.2)	<0.001
Triad of hypoalbuminemia, anemia, and edema at diagnosis (*n*, %)	11 (47.8)	3 (9.1)	<0.001
Infants who had LRTI ^ requiring hospital admission before diagnosis (*n*, %)	16 (69.6)	26 (78.8)	<0.433
PseudoBartter presentation (*n*, %)	1 (4.3)	13 (39.4)	<0.003
Infants who never had an opportunity to receive PERT for >2 weeks	14 (60.9)	0	<0.0001
Sibling death with suspected CF (%)	10 (43.5)	5 (15.2)	0.019

* Fecal elastase level of less than 200 mcg/g of stool or by identifying two biallelic pathogenic CFTR variants associated with pancreatic insufficiency. ** Weight-for-height below −3 SD with the presence of edema. ^#^ Severe symptomatic anemia (<7 gm/dL) needing blood transfusion before diagnosis of CF. ^ Lower respiratory tract infection.

**Table 3 IJNS-11-00067-t003:** Description of univariate risk factor analysis for those who died before 1 year of age.

		Dead ≤ 12 Months *N* (%)	Alive > 12 Months *N* (%)	Odds Ratio	*p* Value
Albumin < 2 g/dL	Yes	8 (47.1 *)	3 (8.3 *)	9.78 (2.14–44.6)	0.002
	No	9 (52.9 *)	33 (91.7 *)		
Severe malnutrition with edema	Yes	9 (50.0)	7 (18.4)	4.43 (1.28–15.23)	0.018
	No	9 (50.0)	31 (81.6)		
PseudoBartter syndrome	Yes	1 (5.6)	13 (34.2)	0.11 (0.01–0.95)	0.044
	No	17 (94.4)	25 (65.8)		
Blood transfusion for severe anemia	Yes	11 (61.1)	13 (34.2)	3.02 (0.95–9.65)	0.062
	No	7 (38.9)	25 (65.8)		
Edema	Yes	8 (44.4)	6 (15.8)	4.26 (1.19–15.3)	0.026
	No	10 (55.6)	32 (84.2)		
Triad of anemia, edema, hypoalbuminemia	Yes	8 (44.4)	6 (15.8)	4.27 (1.19–15.3)	0.026
	No	10 (55.6)	32 (84.2)		
Homozygous *CFTR* pathogenic variants	Yes	15 (83.3)	17 (45.9)	5.88 (1.45–23.8)	0.013
	No	3 (16.7)	20 (54.1)		
Suspected/confirmed diagnosis of CF in sibling	Yes	9 (50.0)	6 (15.8)	5.33 (1.49–18.9)	0.01
	No	9 (50.0)	32 (84.2)		
Meconium Ileus	Yes	1 (5.6)	6 (15.8)	0.31 (0.04–2.82)	0.314
	No	17 (94.4)	32 (84.2)		
PERT for at least 2 weeks	Yes	4 (22.2)	31 (100.0)	Convincing, but cannot be calculated	Highly significant
	No	14 (77.8)	0 (0.0)		
PI	Yes	18 (100.0)	31 (81.6)	Convincing, but cannot be calculated	Highly significant
	No	0 (0.0)	7 (18.4)		

* Serum albumin levels were available for 53 patients.

## Data Availability

The data presented in this study are available on request from the corresponding author due to patient privacy, and legal reasons.

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
