# Peer review of "Exceptionally High Cystic Fibrosis-Related Morbidity and Mortality in Infants and Young Children in India: The Need for Newborn Screening and CF-Specific Capacity Building"

_2409-515X, 2025, doi:10.3390/ijns11030067_

Round 1
Reviewer 1 Report
Comments and Suggestions for Authors
In this study, the authors analyzed 56 infants diagnosed with CF in infancy and evaluated the clinical factors associated with mortality in this population. There was a high level of mortality in this cohort with 41% of the subjects who had died with average death of 5 months. The authors identified several risk factors for mortality, including hypoalbuminemia, Kwashiorkor, anemia and edema. Their findings are very important in identifying the risk of untreated and and late diagnosis of cystic fibrosis and they make a strong argument for the importance of early detection, screening and specifically a clear argument for the benefit of newborn screening. To improve their argument, the authors may want to include mortality rates in infants with CF in European and North American countries where NBS is implemented.
- The authors must be careful to assume the three infants had CF without a formal diagnosis based off of parental carrier status and siblings with CF alone. Can they be more specific when they say that they had "classic CF phenotype"? Did they have evidence of malabsorption? Respiratory symptoms?
- Pseudomonas detection rates were almost 60% for this cohort. Is that typical in India for this age group? I am surprised that there was less S. aureus detection which tends to be the more typical organism in infants. Regarding the Pseudomonas (PA), I would like to have seen the comparative analysis and risk factor analysis tables to include the risk of mortality in those with and without identified Pseudomonas. was this excluded because the vast majority of cultures had PA? It would not be appropriate to assume those without cultures
- The nomenclature for describing the CFTR variants should be consistent between the table and graph. In the table, they use F508del but in the graph they use c.1521_1523delCTT (the latter component, CTT may not be appropriate here).
- For table 3: The title should be a title - remove the word "describes". In addition, I would remove the comment "highly significant" under PERT and PI as that is speculative and not scientifically shown
- Under 4.4. Where is the data that shows that "the strategy of proactively identifying babies who present with dehydration in summer resulted in better survival of those babies"
Comments on the Quality of English Language
They should use consistent spelling of edema (oedema vs edema) throughout the manuscript.
Under 4.1, there are several run on sentences that need some editing to help with understanding (e.g. the sentence stating "In our study cohort, infants")
Under 4.5 "poor weight gains" should be poor weight gain.
Author Response
Please see the attachment. I have replied to both reviewers in this document as there were a few common concerns, and also to clarify with both reviewers why some changes were made.

Reviewer 2 Report
Comments and Suggestions for Authors
Thank you for a very interesting and well-written paper.
Results: I believe the authors' narrow definition of kwashiorkor is overlapping with other outcomes and not so relevant for this study. I recommend omitting this definition and instead using more clear-cut outcomes such as hypoalbuminemia, anemia, and peripheral edema throughout the paper, which also will give . These outcomes are easier to interpret and replicate in future studies. Kwashiorkor is strongly associated with areas having limited access to diverse and protein-rich foods, which is also the case in parts of India.
Discussion: Section 4.2
The authors write: "The clinical triad of severe anemia, hypoalbuminemia, and edema was a significant predictor of mortality (p < 0.001; OR 4.2, 1.19-15.26). Although this combination was described in Danish infants born during 1949-1980, its significance for diagnosis and prognosis has not been recognized previously."
This statement is not correct. Every healthcare professional (HCP) who has worked with cystic fibrosis for some years has seen this either before screening was implemented or in false-negative screening patients. See also this reference from the 1960s: Pittman FE, Denning CR, Barker HG. Albumin Metabolism in Cystic Fibrosis. Am J Dis Child. 1964;108(4):360–365. doi:10.1001/archpedi.1964.02090010362005.
4.7 Newborn Screening
In India, some states and private hospitals have implemented newborn screening (NBS) programs, but many people do not have access. Based on the results of this paper, I would recommend toning down the emphasis on newborn screening. The most important takeaway, in my opinion, is the awareness of cystic fibrosis diagnosis and early treatment. Establishing standards of care for diagnosis and treatment, hygiene and infection control, prevalence overview, and quality control (such as establishing registries) is crucial. There is no use in having a screening program for cystic fibrosis if the most basic treatment algorithms and care standards are not available. This has also been suggested by the authors in other parts of the paper.
Author Response

(The authors gave the same response as above.)

Reviewer 3 Report
Comments and Suggestions for Authors
This is a critically important topic and the manuscript is compelling and very well written.
It is rare that I complete a peer review with so few edits or suggestions. I highlighted the verbiage in the PDF that correspond to the minor feedback below.
|
PDF page and location |
Comment |
|
Page 5, Figure 1 |
Some additional detail might be helpful. With 55 infants with genetically confirmed CF, I’m thinking of 110 alleles (if all infants had both CF-causing variants identified); but the percents depicted in the pie chart (e.g., 26%, 35%, 21%) don’t correspond to a whole number out of 110 (e.g., 26% of 110 alleles would tell me that there were 28.6 F508dels). Perhaps clarify total number of variants identified among the 55 patients, and/or add a decimal to the tenth place on percents. Could you also add a box under “other” that lists the other variants identified and frequency (by count or %)? |
|
Page 6, Table 2 |
“Oedema” is used in the description of the triad in the table, while “edema” is used elsewhere. I recognize dialectical differences, but recommend harmonizing throughout. |
|
Page 6, last sentence of last paragraph |
Says “clinical trial” where it should say “clinical triad” |
|
Page 7, Table 3 |
“PseudoBartter” has no space between the words where in other instances “Pseudo Bartter” is used. |
|
Page 7, Table 3 |
Is this really any homozygous variants? Or homozygous F508del? Or did all homozygotes happen to be homozygous for PI-causing severe variants? It might be worth clarification in the table or text. I would venture it’s not the homozygosity driving the increased mortality as much as bi-allelic class I/II variants (unless children of consanguineous couples are more sick due to genetic modifiers where there’s more loss of heterozygosity across the genome…). |
|
Page 7, First paragraph of discussion |
Looks like punctuation is missing between the sentence ending with “Kwashiorkor” and the next sentence (add period, capitalize “The WFA…”, if I’m reading the sentence breakpoint correctly). |
|
Page 9, first sentence |
“On” should be “of”, and the conjunctive sentence should be edited slightly, potentially to read “Our data support the imperative need for education and awareness of CF among community caregivers and general pediatricians, emphasizing careful attention to poor weight gains….” |
|
Page 9, second paragraph of Misdiagnosis |
“Compounded” should be “confounded” |
|
Page 9, second paragraph of Misdiagnosis |
The highlighted sentence “Routine implementation of NBS programs in India…” might be better with edits, as the beginning of the sentence seems to speak to NBS at-large (for multiple conditions, programmatic level), and the second part of the sentence links these large-scale challenges to CF-specific problems. It might be just saying “CF NBS” instead of “NBS programs” in this instance, since the next section addresses more systemic challenges of NBS implementation. |
|
Page 10, last sentences of top paragraph |
For continuity (what we have now vs. what we hope for) I might clarify that the proposed clinical tool offers an immediate improvement to no screening (while it can eventually help ID false negatives after NBS, that’s not the current state). Something like “Until NBS is standardized, our proposed clinical tool may facilitate recognition of life-threatening CF, and continued use after NBS implementation may recognize critically ill infants with false negative results.”
Recommend removing “But” from the beginning of the last sentence of paragraph (rest of sentence is good). |
|
Page 10, first sentence of conclusions |
One participant did not have genetic confirmation – right? Previously stated inclusion was diagnosis by sweat and/or genetics, with 1/56 missing genetics. So I think this sentence should be changed to remove “by genetic confirmation”. |
|
Page 10, second sentence of conclusions |
“Improved” should be “improve” |

Author Response
Please see attached Redlined version of the Manuscript. Thank you.

Round 2
Reviewer 2 Report
Comments and Suggestions for Authors
The abstract in the submission system should be corrected, so it aligns the latest version of the manuscript.
I have no further comments.